# Is the Side Bridge Test Valid and Reliable for Assessing Trunk Lateral Flexor Endurance in Recreational Female Athletes?

**DOI:** 10.3390/biology11071043

**Published:** 2022-07-12

**Authors:** Casto Juan-Recio, Amaya Prat-Luri, Alberto Galindo, Agustín Manresa-Rocamora, David Barbado, Francisco J. Vera-Garcia

**Affiliations:** 1Sports Research Centre, Department of Sport Sciences, Miguel Hernández University of Elche, Avda. de la Universidad s/n, 03202 Alicante, Spain; cjuan@umh.es (C.J.-R.); alberto.galindo@goumh.umh.es (A.G.); amanresa@umh.es (A.M.-R.); dbarbado@umh.es (D.B.); fvera@umh.es (F.J.V.-G.); 2Institute for Health and Biomedical Research (ISABIAL Foundation), Miguel Hernández University of Elche, Avda Pintor Baeza, 12—Planta 5^a^ Centro de Diagnóstico, 03010 Alicante, Spain

**Keywords:** field test, core, shoulder, electromyography, muscle fatigue, anthropometry, consistency

## Abstract

**Simple Summary:**

Although the side bridge test has been widely used for assessing trunk lateral flexor endurance in sport, clinical, and scientific settings, to the best of the authors’ knowledge, no study has analyzed its validity and reliability in an only female population. The surface electromyography of eight abdominal, back, shoulder and hip muscles was measured during the test to analyze its validity. A one-week test-retest design was performed to evaluate its reliability. No significant differences were found between the trunk lateral flexors and the deltoids. The study data showed that the test performance could be significantly predicted by external oblique and deltoid normalized median frequency slopes and by body mass and trunk height. Based on the results of this study, the shoulder muscle activation and fatigue and the individuals’ anthropometric characteristics, especially the mass, played an important role in the side bridge test performance, which questions the validity of this multi-joint test to specifically assess trunk lateral flexor endurance. In addition, although the side bridge test showed a good data consistency, its intra-subject variability was high, which reduces its utility when small intra-subject changes in muscle endurance are important (e.g., elite sport).

**Abstract:**

The side bridge test (SBT) is one of the most popular tests to assess isometric trunk lateral flexor endurance. The aim of this study was to assess the validity and reliability of the SBT in healthy females. Twenty-four (24.58 ± 3.92 years) physically active (1–2 h of moderate physical activity, 2–3 times a week) females voluntarily participated in this study. The surface electromyography (EMG) of eight abdominal, back, shoulder and hip muscles was measured during the SBT. Normalized median frequency slopes (NMF_slope_) were calculated to analyze the muscle fatigue. The EMG amplitudes were normalized to maximum EMG values to assess muscle activity intensity. A one-week test-retest design was performed to evaluate the SBT reliability through the ICC_3,1_ and typical error. Higher NMF_slopes_ and normalized EMG amplitudes were found in deltoids, abdominal obliques, rectus abdominis, and erector spinae in comparison to latissimus dorsi, gluteus medius, and rectus femoris. However, no significant differences were found between the trunk lateral flexors and the deltoids. Linear regression analysis showed that SBT performance could be significantly predicted by external oblique and deltoid NMF_slope_ (adjusted R^2^ = 0.673) and by body mass and trunk height (adjusted R^2^ = 0.223). Consistency analysis showed a high intraclass correlation coefficient (0.81) and a relatively high typical error (10.95 s). Despite the good relative reliability of the SBT, its absolute reliability was low and its validity questionable, as the shoulder muscle activation and fatigue and the individuals’ anthropometric characteristics played an important role in SBT performance.

## 1. Introduction

Numerous field tests have been used to evaluate trunk muscle capacity in sport and clinical settings [1,2,3] because of its relation to sport performance and low back pain [4,5]. The side bridge test (SBT) is one of the most widely used standardized field-based tests for assessing the trunk lateral flexor endurance in scientific, sport, and clinical settings [2,5,6,7]. This test basically consists of maintaining a lateral lying position (against gravity) supported by the elbow-forearm and feet for as long as possible [2,7]. The main reasons for its popularity are its high relative consistency [intraclass correlation coefficient (ICC) > 0.75) [2,5,6,7], that it requires minimal and inexpensive equipment, and its safety and ease of use.

However, although the SBT is generally considered a trunk endurance test, in the last years several studies have reported that some individuals prematurely ended the SBT because of upper extremity fatigue or pain [8,9,10]. Concretely, Greene et al. (2012) [9] showed that the 42.5% of participants reported upper extremity fatigue or pain as the reasons for ending the test versus the 45.8% who reported trunk side or hip fatigue or pain. Likewise, Roth et al. (2016) [10] showed that the 59% of individuals reported upper or lower extremity fatigue after the SBT performance compared to the 23% who reported trunk fatigue. Furthermore, some authors who analyzed a similar test, such as the prone bridging test, highlighted the importance of upper body strength to assume the bridging position, and muscular endurance to sustain it [11]. In this sense, a relationship has been found between interscapular muscle endurance and SBT performance [12]. Therefore, considering that in the SBT the individuals are only supported on one side, the upper extremity condition could be a limiting factor in test performance, especially in participants with shoulder muscle weakness and/or pain [9,13]. Besides, some authors have indicated that the participants’ anthropometric characteristics (i.e., mass, height, etc.) could also have an influence on the SBT score [7,14,15] and especially the body mass that is not supported on the mat during the test execution.

Despite all the factors that could affect SBT performance, there is a lack of studies analyzing the validity of this multi-joint test for specifically measuring trunk lateral flexor endurance. On the other hand, some studies have analyzed the validity of several trunk extensor and flexor muscle endurance field tests (i.e., Biering-Sorensen test, prone bridge test, flexor endurance test) by analyzing trunk and limb muscle fatigue and its relationship to test performance. The frequency spectrum of the surface electromyography (EMG) has been used to analyze the muscle fatigue, as it causes a decline of the frequency content of the EMG signal, usually described as a decrement of the median frequency parameters of the EMG spectrum [16,17,18]. Although there are no EMG studies describing the validity of the SBT, different authors have analyzed muscle activation during the side bridge exercise (in which the same posture is maintained) [19,20,21]. Most of these EMG studies have focused on the analysis of the trunk muscle activation [20,21,22], but very few have also analyzed other muscle groups that could also play an important role in exercise performance. In this sense, although gluteus medius activations of 74 ± 30% of the maximum voluntary isometric contraction (MVIC) have been reported during the execution of the side bridge exercise [19], to the best of the authors’ knowledge no studies have analyzed the shoulder muscle activation while maintaining this one-sided position.

Therefore, considering that both the hip and shoulder muscle activation and the participant’s anthropometric characteristics (i.e., mass, height, leg length, etc.) could have a significant effect on SBT performance, the aim of this study was to analyze the validity of this test for measuring trunk lateral flexor endurance. Specifically, the amplitude and median frequency characteristics of the EMG signals recorded from different abdominal, back, shoulder and hip muscles of healthy recreational female athletes were assessed to investigate the influence of these muscles’ activity intensity and fatigue on SBT performance. In addition, the relationships between some participants’ anthropometric characteristics, muscle fatigue and SBT performance were also analyzed. Moreover, given that most of the SBT reliability studies have been conducted with males and that the few studies that have analyzed the absolute reliability have shown questionable values [standard error of measurement = 15–20%] [5,7], the absolute and relative reliability of this test was analyzed in the female population.

## 2. Methods

### 2.1. Participants

A total of twenty-four (24.58 ± 3.92 years; 60.90 ± 2.90 kg; 163.49 ± 5.60 cm) physically active (1–2 h of moderate physical activity, 2–3 times a week) females voluntarily participated in this study. The inclusion criteria for taking part in the study were: (i) being a woman; (ii) not participating in trunk exercise programs at the time of the study; (iii) not having any known medical problem; and (iv) not having had episodes of back, hip or shoulder pain in the 6 months before the study. The participants were asked to sign an informed consent approved by the University Office for Research Ethics (DPS.RRV.05.15) according to the Declaration of Helsinki.

The sample size used in this study was previously estimated with the sampling software package, GPower 3.1. Given that the Side Bride performance was expected to be significantly associated with the decline in the EMG frequency of no more than five muscles (external and internal oblique muscles, deltoid, erector spinae and gluteus medius), a sample of 24 participants was needed to detect a significant large effect size (R^2^ = 0.36; f^2^ = 0.75; power = 80%; α = 0.05) on a multiple linear regression model with five potential significant predictors.

### 2.2. Side Bridge Test

The participants were placed in a lateral decubitus position supported on their preferred forearm and elbow, with their shoulder and elbow in a 90° flexion, and their legs extended and barefoot while maintaining the alignment of the body segments forming a straight line between their shoulder, hip, and feet (Figure 1). The foot of their non-preferred leg was positioned in front of the foot of their preferred leg and the hand of their free arm was placed on their contralateral shoulder. The test consisted of maintaining the aforementioned position for as long as possible until exhaustion while the examiner provided verbal feedback of the position and vigorously encouraged them [2,7]. The endurance time was recorded manually using a digital chronometer (Casio HS-30W-N1V, Tokyo, Japan) until the participant gave up or was not able to maintain the correct position for more than 3 s.

### 2.3. Procedures

In order to analyze the test-retest reliability, participants performed the SBT in two testing sessions carried out in a biomechanics lab one week apart from each other. Two experienced examiners supervised the two testing sessions to ensure standardized testing procedures. In the first session, the participants filled out a questionnaire about their medical history and sport practice in order to know their health status and level of physical activity. Subsequently, the following anthropometric measurements were carried out: mass, height, siting height, trunk height, biacromial diameter (distance between the two acromial processes), bicrestal diameter (distance between the two anterior-superior iliac spines) and acromial-iliac index (bicrestal diameter divided by biacromial diameter × 100). A 5-min standardized warm-up protocol was performed before the SBT in both sessions. This protocol consisted in the following exercises: pelvic circumductions (5 repetition for each direction), pelvic retroversions (5 repetitions), pelvic anteversions (5 repetitions), cat-camel exercise (10 repetitions), crunches (10 repetitions), extensions in prone position on a stretcher (10 repetitions), frontal bridge (15 s), side bridge (15 s each side), and back bridge (15 s). The EMG activity was recorded during the SBT execution of the second testing session to analyze the SBT validity.

### 2.4. Equipment, Data Registration, and Signal Processing

The EMG signals were recorded at 1000 Hz with an 8-channel surface EMG system (Muscle Tester ME6000^®^; Mega Electronics Ltd., Kuopia, Finland). EMG data were amplified with an analog differential amplifier (overall gain of 1000 and a common mode rejection ratio of 110 dB), analogue-to-digital converted (14-bit) and filtered using a Butterworth band pass of 8–500 Hz (−3 dB points). The input impedance was 10 GΩ.

Topographic marking was performed by the palpation of different anatomical points to facilitate the electrode placement. Then, the skin was shaved and cleaned with alcohol to decrease the electrode-skin impedance. After allowing the skin to dry, two Ag-AgCl surface electrodes (Arbo Infant Electrodes, Tyco Healthcare, Neustadt, Germany) were placed on the following muscles and locations of the participants’ preferred side [23,24,25]: (i) rectus abdominis (RA), approximately 3 cm lateral to the umbilicus; (ii) external oblique (EO), at the intersection of the imaginary line joining the anterior-superior iliac spine and the costal angle with the imaginary line running transverse to the spine from the center of the umbilicus; (iii) internal oblique (IO), the geometric center of the triangle formed by the inguinal ligament, the outer edge of the rectus sheath and the imaginary line joining the anterior-superior iliac spine and the umbilicus; (iv) rectus femoris (RF), halfway between the anterior-superior iliac spine and the patella; (v) gluteus medius (GM), halfway on the line joining the iliac crest to the trochanter of the femur; (vi) deltoids (DE), on the largest bump on the imaginary line from the acromion to the lateral epicondyle of the elbow; (vii) latissimus dorsi (LD), 4 cm below the lower tip of the scapula over the belly muscle; (viii) erector spinae (ES), 3 cm lateral to the spinous process of L3. The pair of electrodes were placed within the borders of the muscles with a parallel orientation to the muscle fibers and with a distance of 25 mm between the center of each electrode. Finally, the electrodes were fixed with tape on their non-metallic part to ensure that they were well fixed to the skin and a mesh was placed on the trunk to minimize the movement of the electrodes and the wiring while testing.

Two repetitions of MVIC against manual resistance were performed prior to the recording of the SBT performance to obtain reference values to normalize the EMG of the aforementioned muscles. Based on the protocol of Vera-Garcia et al. (2010) [26] maximal isometric trunk flexions, right and left trunk rotations, and right and left lateral trunk bends were performed for RA, IO, and EO, and maximal isometric trunk extensions for ES. In addition, following the protocols of Konrad et al. (2001) [27] and Thorborg et al. (2010) [28] maximal isometric knee extensions and hip abductions were performed for RF and GM, respectively. To obtain the MVIC of DE, participants laid supine with their dominant side at the edge of a stretcher, with their shoulder at a 90° abduction and the elbow at a 90° flexion. The examiner held them at the elbow and wrist by exercising resistance while the participants tried to abduct the shoulder. To obtain the MVIC of LD, two maneuvers were performed: (i) the participants were placed in prone position with their dominant side at the edge of the stretcher, positioning their shoulder in external rotation at a 90° abduction and their elbow at a 90° flexion. The examiner held the stretcher and was positioned so that his hip immobilized the participant to prevent her from sliding when she tried to perform a pull-up; (ii) the participant was placed in the same position but with her shoulder in a neutral position. Once in this position, she performed the rowing exercise in a vertical direction while the examiner exerted manual resistance in the opposite direction. Another examiner held the participant’s trunk to avoid any movement. Each MVIC was maintained for 5 s, and a 3 min rest was allowed between sets. X was prevented from moving while immobilized.

### 2.5. Data Analysis

After visual examination to eliminate possible artifacts, the raw EMG signals were analyzed in the time and frequency domain. In order to compare the amplitude of the EMG signal in the SBT between the different muscles, the raw data were full-wave rectified, smoothed using a moving average window of 100 ms and subsequently, normalized to maximum EMG values obtained during the MVIC. The median frequency (MF) of the EMG power spectrum was calculated in each 1-s interval of raw EMG data with fast Fourier transformation with Megawin software v3.0 (Mega Electronics Ltd., Kuopio, Finland). The MF was defined as the frequency that divides the area of the calculated spectrum in half. Fatigue causes a decrease of the frequency content of the EMG signal (i.e., a decline of the MF parameters of the EMG power spectrum). Therefore, a linear regression analysis was applied to the MF time series (MF as a function of time) to estimate the degree of decay (i.e., the MF_slope_), which represents muscle fatigue. MF_slopes_ were divided by the initial MF and multiplied by 100 (%/s) to express the decline rate of MF as the percent change from the initial value (NMF_slopes_) [16,17].

### 2.6. Statistical Analysis

The normal distribution of all the data series was verified with the Shapiro-Wilk test (*p* > 0.05). Subsequently, descriptive statistics (mean and standard deviation) were calculated for each of the variables. To analyze the absolute inter-session reliability of the SBT, the typical error (% intrasubject variation) and its 95% confidence limits and the coefficient of variation (CV) were analyzed. The typical error was calculated as the standard deviation of the difference between session 1 and 2 divided by √2 and the CV was calculated as the standard deviation divided by the mean. In addition, the relative reliability was analyzed with the intraclass correlation coefficient (ICC_3,1_), calculating its 95% confidence limits. Thus, based on the method previously described by Hopkins, (2000) [29], the ICC was calculated from the analysis of variance: (F − 1)/(F + k + 1), in which F is the F-ratio of the subject and k is the number of trials (k = 2). The interpretation of the ICC was made based on the following values: excellent (0.90–1.00), good (0.70–0.89), moderate (0.50–0.69) and low (<0.50) [30]. Finally, the change in the mean was also calculated and a t-test analysis was conducted to analyze the systematic error.

The following methods were used to examine the SBT validity based on the EMG amplitude and the MF variables obtained in the second testing session. First, a one-way repeated measures ANOVA design (Huynh-Feldt correction) was conducted to investigate the main effect of the independent variable muscle on the dependent variable. To investigate which muscles are relatively more active during the test, post-hoc comparisons with Bonferroni correction were performed to compare the means of the normalized EMG amplitude data. Then, another ANOVA of repeated measures (Huynh-Feldt correction) was used to analyze the differences between the NMF_slopes_ of the muscles, and post-hoc pairwise comparisons were made with Bonferroni correction to compare the mean NMF_slopes_ of each muscle with each other.

In addition, to further explore the SBT validity, Pearson correlation coefficients (*r*) were calculated between the SBT endurance times (obtained in the second testing session) and both the anthropometric variables and the NMF_slopes_ of the different muscles. The interpretation of the *r* was set in accordance with Cohen [31] as low = 0.10–0.30, moderate = 0.30–0.50, and high > 0.50. Finally, multiple backward linear regression analyses were performed to assess which NMF_slope_ and anthropometric variable best predicted the endurance time. The null hypothesis was rejected at the 95% significance level (*p* ≤ 0.05). All the statistical analyses were performed with SPSS v23.0 software (SPSS Inc., Chicago, IL, USA).

## 3. Results

Descriptive statistics of the participants’ anthropometric characteristics are shown in Table 1.

### 3.1. Reliability

Descriptive statistics and reliability analysis of the SBT are shown in Table 2. The mean endurance time values for session 1 and 2 were 77.25 ± 26.62 s and 68.96 ± 21.21 s respectively, showing a significant decrease between both sessions (t = 2.624; *p* = 0.015). The endurance time varied greatly between participants (from 38 to 132 s). The relative reliability was good (ICC = 0.81), and the absolute reliability was low (typical error = 10.95 s; coefficient of variation = 30.75%).

### 3.2. Validity

#### 3.2.1. Differences in Normalized EMG Amplitude between Muscles

The Huynh-Feldt correction in the one-way repeated measures ANOVA (sphericity was not assumed) showed a significant difference in normalized EMG amplitude between the different muscles during the SBT (F = 21.274; *p* < 0.001). The mean and standard deviations for the normalized EMG amplitude of the trunk, hip and shoulder muscles are shown in Figure 2. The pairwise comparison with Bonferroni correction showed that the normalized EMG amplitudes of EO, RA, IO, DE and ES were higher than those of LD and RF (*p* < 0.05), which were the muscles that obtained the lowest mean activation levels (<15% MVIC). On the other hand, OE and RA showed the highest activation levels (50.2% and 41.9% MVIC), being significantly higher than those of the GM. Besides, the GM showed a significantly greater activation than RF.

#### 3.2.2. Differences in NMF_slope_ Values between Muscles

The one-way repeated measures ANOVA (Huynh-Feldt correction) found significant differences in the NMF_slope_ between muscles during the SBT (F = 6.764; *p* < 0.001). The mean and standard deviations of the NMF_slope_ values for the trunk, hip, and shoulder muscles are shown in Figure 3. Post-hoc Bonferroni comparisons showed significant differences between DE (which showed the greatest decrease in NMF_slope_) and LD, GM, and RF (*p* < 0.05). Additionally, EO showed a higher decrease in NMF_slope_ than LD.

#### 3.2.3. Correlation Coefficients between NMF_slope_ Values and Endurance Times

The Pearson correlation analysis between the SBT endurance time and the NMF_slope_ of the muscles (Table 3) showed high significant correlations for DE, EO, and ES (*r* ≥ 0.650). On the other hand, the moderate correlations obtained for IO, RA, LD, and GM (0.386 ≤ *r* ≤ 0.235) and the low correlation showed for RF (*r* = 0.036) were non-significant.

Multiple backward linear regression analyses, with all NMF_slope_ values as independent variables and SBT endurance time as dependent variable, revealed that the best model to predict the endurance time should include the NMF_slope_ of EO and DE (*p* < 0.05). The resulting regression equation could be written as: endurance time = 102.120 + 55.434 × NMF_slope_ of EO + 25.832 × NMF_slope_ of DE (adjusted R^2^ = 0.673).

#### 3.2.4. Correlation Coefficients between Anthropometric Variables and Endurance Times

The Pearson correlation analysis between the SBT endurance time and the anthropometric variables (Table 4) showed a significant moderate correlation of *r* = −0.416 for the body mass. On the other hand, the moderate correlations of the endurance time with the participant’s height (*r* = −0.361) and trunk height (*r* = −0.371) were quasi-significative (*p* = 0.083 and 0.074, respectively). Finally, the moderate-to-low correlations obtained for sitting height, biacromial diameter, and bicrestal diameter (*r* ≤ −0.312) were non-significant.

Multiple backward linear regression analyses, with all anthropometric variables as independent variables and SBT endurance time as dependent variable, revealed that the endurance time could be significantly predicted by body mass and trunk height (*p* < 0.05). The resulting regression equation could be written as: endurance time = 247.773 − 1.159 × mass − 2.168 × trunk height (adjusted R^2^ = 0.223).

## 4. Discussion

The present study aimed to explore the SBT characteristics by analyzing the muscles’ activity intensity and fatigue in the test performance, the influence of the participants’ morphological characteristics, and the test-retest reliability. The main findings showed the important role of deltoids and the influence of anthropometric characteristics, especially the mass, on SBT performance. These results question the validity of this multi-joint test for specifically assessing trunk lateral flexor endurance in physically active females and could contribute to a better interpretation of the SBT scores.

The mean SBT endurance time recorded in this study (69.0 ± 21.2 s) was very similar to that obtained by McGill et al. (1999) [2] and Greene et al. (2012) [9] in female university students (72 ± 31 s and 71.3 ± 31.3 s, respectively), and lower than that obtained by Evans et al. (2007) [5] in elite (state level) female athletes (91.1 ± 38 s). With regard to the consistency analysis (Table 2), the SBT showed a good relative reliability with an ICC = 0.81, which is very similar to that obtained in previous studies [5,9]. However, the typical error was relatively high (10.95 s; 7.48%), which is in line with prior studies that have analyzed the absolute reliability of this and other trunk endurance field tests in females [5] and males [5,7,32]. It seems that a high intra-subject variability is a common feature of these tests, which reduces their ability to detect small changes in trunk endurance [5,32]. In addition, it must be noted that there was a significant decrease in SBT endurance time in the second session compared to the first session, possibly related to the participant demotivation by the extended testing time of the second session, in which the EMG protocol was carried out (i.e., electrode site location, skin preparation, electrode placement, MVIC practice and recording, etc.).

The normalized EMG amplitude data showed a significantly higher activation of the abdominal wall (40.6% < MVIC < 50.2%), DE (36.3% MVIC) and ES (28.2% MVIC) than LD (14.0% MVIC) and RF (5.85% MVIC) during the SBT (Figure 2). Previous studies which analyzed the side bridge exercise described similar trunk activation levels to those obtained in this study [19,23,33,34,35], showing the importance of the oblique abdominal muscles to generate trunk lateral flexion torques. Interestingly, the DE activation levels found in this study did not differ significantly from those obtained by the trunk lateral flexors, highlighting the DE relevance for maintaining the shoulder abduction during the side bridge position. On the other hand, despite the fact that the GM is an important hip abductor, it showed a mean activation level of only 21.2% MVIC, which was significantly lower than that of the EO and RA. Surprisingly, a previous study performed by Ekstrom et al. (2007) [19] found a GM activation level of 74% MVIC during the side bridge exercise. The different results obtained in both studies could be explained by differences in side bridge execution (e.g., different duration, different placement of the upper arm, etc.), in EMG signal processing and analysis (e.g., different MVIC techniques and normalization procedures for GM) and/or in participant characteristics (e.g., different participant age and physical condition).

The NMF_slope_ results were in line with those of the normalized EMG amplitude, as there were no statistical differences in the MF decline (i.e., muscle fatigue) during the SBT between DE, EO, IO, RA, and ES. It is noteworthy that the DE was the muscle that showed the highest MF drop (−0.50%/s) throughout the test, finding significant differences with respect to LD, GM and RF (Figure 3). In addition, the correlation analysis showed significant high correlations between the SBT endurance time and the NMF_slope_ of EO (*r* = 0.722), DE (*r* = 0.650) and ES (*r* = 0.664). Previous studies analyzing the validity of other isometric trunk endurance tests (i.e., Biering-Sorenesen test, prone bridge test, and flexor endurance test) have also reported significant correlations between endurance times and MF declines of the trunk and hip muscles, showing the global nature of these multi-joint tests [16,17,18] Moreover, a multiple backward linear regression was performed in this study to determine which muscle best predicted the SBT endurance time, finding that OE and DE fatigue significantly predicted the SBT endurance time with an adjusted R^2^ = 0.673. Overall, the current EMG results highlight the importance of both the shoulder and trunk muscles in maintaining the side bridge posture during the test, and support previous studies that have reported that upper extremity fatigue/pain are one of the causes of SBT ending [8,9,10]. Given the influence of the upper extremities in conventional SBT performance, some studies have proposed alternative tests to minimize or eliminate its influence, as for example the modified SBT (with a 30 or 45° of trunk inclination on a roman chair) and the feet-elevated side support [9,36,37]. However, they have not become as popular as the SBT, perhaps because some of them need equipment for their execution (i.e., Roman chair). Future studies should analyze the validity of these alternatives for specifically assessing trunk lateral flexor endurance.

With regard to the correlation analysis between the participants’ anthropometric characteristics and SBT endurance times (Table 4), a moderate significant negative correlation was found for participants’ body mass (*r* = −0.416). This finding is in line with previous studies which have also found significant correlations between body mass and the endurance times of this (r = −0.610) [7] and other similar trunk endurance tests (−0.29 ≤ *r* ≤ −0.39) [14,15,38]. Moreover, the multiple backward linear regression performed in this study showed that the participants’ mass and trunk height significantly predicted the SBT performance, with an adjusted R^2^ = 0.223. Therefore, the participants’ mass seems an important factor for the performance of these holding tests, in which the participants have to maintain most of their body mass raised against gravity until failure. In addition, a higher height or a higher trunk height (which obtained quasi-significant negative correlations with SBT endurance time) also seem to be a disadvantage for the SBT execution, as the body support points (forearm-elbow and feet) are further apart, which increases the lever arm of the body weight. Unlike our study, a previous study performed in males found significant correlations between the bicrestal and biacromial diameters and the SBT endurance time, which highlights the importance of body mass distribution [7]. Although further research is needed to understand the influence of these and other anthropometric variables on SBT performance in different male and female populations better, these results indicate that they have an impact on SBT endurance time, which must be taken into consideration when comparing the SBT performance of participants with different morphological characteristics. In these cases, the SBT does not allow a valid comparison, since the possible differences in SBT endurance time between participants could be influenced by participants’ anthropometric differences in mass, trunk height, etc.

### Study Limitations

Several limitations exist as to the interpretation of the data in this study. As usual in EMG studies, variability of normalized EMG amplitude and NMF_slope_ between participants was high (Figure 2 and Figure 3). Although two trained experimenters supervised participants’ SBT execution, small differences in body position during the test, along with differences in physical fitness and sport practice between participants, could explain this variability. In addition, despite the fact that electrode sites were carefully determined to ensure a clear representation of each individual muscle (based on the SENIAM guidelines and previous EMG studies) [23,26,27,28], EMG crosstalk could affect the EMG signals (mainly in IO and EO, lying atop of each other in the anterior-lateral abdominal wall). Finally, interpretation of this study data is limited to our participants being young healthy physically active females. Future studies should analyze the SBT validity in other populations, such as sedentary males and females, athletes of different sports, patients with different spinal conditions, and so on.

## 5. Conclusions

Based on the results of this study, the shoulder muscle activation and fatigue and the individuals’ anthropometric characteristics, specially the mass, played an important role in SBT performance, which questions the validity of this multi-joint test for specifically assessing trunk lateral flexor endurance. Comparisons of SBT results between groups should consider the body weight in their analysis (e.g., using a regression analysis adjusted for body weight or ANCOVA with body weight as a confounding variable). In addition, although the SBT showed a good relative reliability, its absolute reliability was low, which reduces its utility when small intra-subject changes in muscle endurance are important (e.g., elite sport).

## 6. Perspective

Although SBT has been widely used for assessing trunk lateral flexor endurance in sport, clinical and scientific settings [2,5,7], to the best of the authors’ knowledge, this is the first study that has analyzed its validity and reliability in a female population. As is common in trunk endurance tests until failure [5,7], the SBT relative reliability was good, but its absolute reliability was low. Therefore, although the SBT seems able to accurately discriminate differences in endurance time between females with similar characteristics [e.g., to classify a group of classmates in a high school), it may have problems to detect low intra-subject changes after training or rehabilitation, which are usual in highly trained athletes. Interestingly, this study showed the important role of deltoids and the influence of anthropometric characteristics (i.e., mass, trunk height) on SBT performance, which suggests that this multi-joint test is not valid to specifically measure trunk lateral flexor endurance in females with different shoulder muscle conditions and/or with anthropometric differences. Therefore, it would be advisable to develop some type of SBT score normalization (e.g., dividing the endurance time by mass) to reduce the effect of the anthropometric between subject differences, as well as to explore the utility of other similar protocols in which the influence of the upper extremity is minimized or eliminated (e.g., modified SBT on a roman chair, feet-elevated side support, etc.) [9,36,37]. Overall, the findings in this study could contribute to a better understanding and application of the SBT scores and to improve the decision-making process when selecting a test to assess trunk muscle endurance.

## Figures and Tables

**Figure 1 biology-11-01043-f001:**
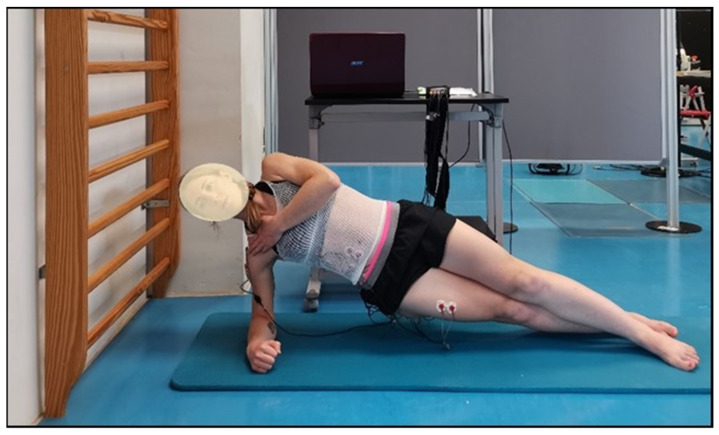
Participant performing the side bridge test on her right side.

**Figure 2 biology-11-01043-f002:**
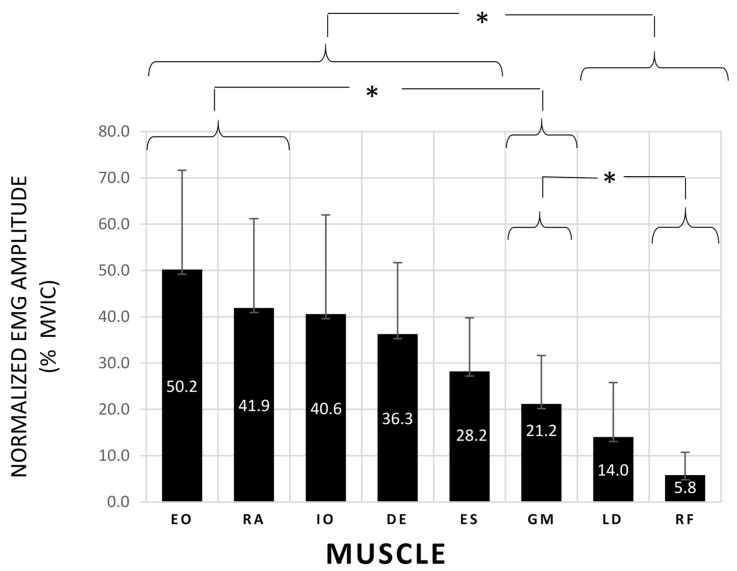
Normalized EMG amplitude obtained during the side bridge test for the following muscles: EO: external oblique; RA: rectus abdominis; IO: internal oblique; DE: deltoids; ES: erector spinae; GM: gluteus medius; LD: latissimus dorsi; RF: rectus femoris. Error bars indicate the standard deviations. * Significant differences after post-hoc pairwise comparisons with Bonferroni correction (*p* < 0.05).

**Figure 3 biology-11-01043-f003:**
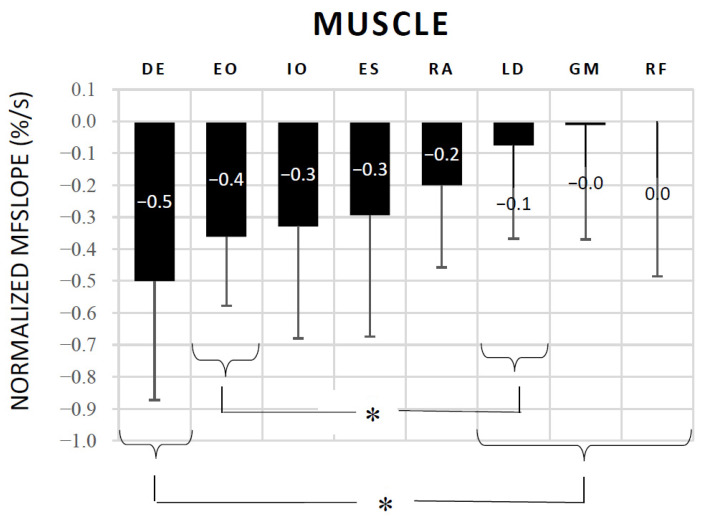
NMFslope obtained during the side bridge test for the following muscles: EO: external oblique; RA: rectus abdominis; IO: internal oblique; DE: deltoids; ES: erector spinae; GM: gluteus medius; LD: latissimus dorsi; RF: rectus femoris. Error bars indicate the standard deviations. * Significant differences after post-hoc pairwise comparisons with Bonferroni correction (*p* < 0.05).

**Table 1 biology-11-01043-t001:** Participants’ anthropometric characteristics.

Mass (kg)	Height (cm)	Trunk Height (cm)	Sitting Height (cm)	Biacromial Diameter (cm)	Bicrestal Diameter (cm)	Acromial-Iliac Index (%)
60.90 ± 6.94	163.49 ± 5.60	49.78 ± 3.53	86.70 ± 2.69	35.89 ± 1.85	27.25 ± 1.89	76.19 ± 5.76

**Table 2 biology-11-01043-t002:** Inter-session reliability of the side bridge test (SBT) endurance time.

	Session 1 (Mean ± SD)	Session 2 (Mean ± SD)	Range	Change in the Mean (95% CL)	Typical Error (95% CL)	ICC_(3,1)_ (95% CL)	%CV
SBT (s)	77.25 ± 26.62	68.96 ± 21.21 *	38–132	−8.29	10.95	0.81	30.75
				(−14.83, −1.75)	(8.51, 15.36)	(0.60, 0.91)	

ICC: intraclass correlation coefficient; CL: confidence limits; CV: coefficient of variation; * Significant with respect to session 1.

**Table 3 biology-11-01043-t003:** Correlations between the normalized median frequency slope (NMF_slope_) of each muscle and the side bridge test endurance time (SBT).

	NMF_slope_ (%/s)							
	EO	ES	DE	RA	IO	GM	LD	RF
SBT (s)	0.722 **	0.664 **	0.650 **	0.403	0.316	−0.311	0.235	0.033

EO: external oblique; ES: erector spinae; DE: deltoid; RA: rectus abdominis; IO: internal oblique; GM: gluteus medius; LD: latissimus dorsi; RF: rectus femoris; Significant correlation: ** *p* < 0.01.

**Table 4 biology-11-01043-t004:** Correlations between the anthropometric variables and the side bridge test endurance time (SBT).

	Mass (cm)	Trunk Height (cm)	Height (cm)	Sitting Height (cm)	Biacromial Diameter (cm)	Bicrestal Diameter (cm)	Acromial-Iliac Index (%)
SBT (s)	−0.416 *	−0.371	−0.361	−0.312	−0.235	−0.188	−0.114

Significant correlation: * *p* < 0.05.

## Data Availability

Not applicable.

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
