# Peer review of "Is the Side Bridge Test Valid and Reliable for Assessing Trunk Lateral Flexor Endurance in Recreational Female Athletes?"

_biology, 2022, doi:10.3390/biology11071043_

Round 1

Reviewer 1 Report

First, I would like to commend the authors for their work in putting this paper together. As mentioned, the SBT is a common test utilized by physical therapists, athletic trainers, fitness specialists, and other practitioners to assess trunk muscular endurance. However, the validity of the test using EMG is not widely known. Though this is a simple study on the surface, it holds novelty and wide-reaching significance. With that being said, I have only a couple of comments:

1) Could you include a power analysis to justify the sample size? With the study recruiting healthy young females, only requiring two visits, and using non-invasive and less physically strenuous procedures, I feel that a higher number of participants could have been acquired. 

2) The first paragraph in the discussion is redundant since you said all this in the introduction already. Cut down on the required reading for your audience. Instead, restate the hypothesis and say whether or not you were correct in your assumptions.

Author Response

We would like to thank you for your advice and recommendations, which in our view have contributed to improving the paper. As you will see, we have performed a revision of this manuscript based on your suggestions.

Below we respond to each comment on a point-by-point basis. The authors’ responses to each comment are in bold,

Reviewer 1

1) Could you include a power analysis to justify the sample size? With the study recruiting healthy young females, only requiring two visits, and using non-invasive and less physically strenuous procedures, I feel that a higher number of participants could have been acquired. 

Response: Thank you for this comment because our sample size selection was actually chosen in a sample size estimation based on previous recommendations. Even though some authors have pointed out that multiple regression analyses require an absolute minimum ratio of 5 cases per significant predictor variable to ensure adequate power and generalizability (Hair et al., 1998; Tabachnick & Fidel~ 1989), it is also true that small sample sizes are prone to high inaccuracy (e.g., large 95% confidence interval). Based on that, we used the sampling software package, GPower 3.1. to calculate the minimum sample size needed to detect significant results in the multiple regression analysis. Considering that the Side Bride performance was expected to be significantly associated with the decline in the EMG frequency of no more than five muscles (mainly oblique muscles, anterior deltoid, erector spinae and gluteus medius), a sample of 24 participants was needed to detect a significant large effect size (R2 = 36%; f2 = 0.75; power = 80%; α = 0.05) on a multiple linear regression model with those five potential significant predictors.

2) The first paragraph in the discussion is redundant since you said all this in the introduction already. Cut down on the required reading for your audience. Instead, restate the hypothesis and say whether or not you were correct in your assumptions.

Response: We agree with you that the first paragraph in the discussion is redundant. We have summarized it including only a reminder of the objective and the main findings.

Reviewer 2 Report

Dear authors and editor, firstly, thanks for the opportunity of reviewing this manuscript.

As I remember, this is the first manuscript that I don't have any comments, suggestions, or doubts for an increase in the quality or clarification. So, I just congrats the authors for the great work and possible publication in this excellent journal (it's important to remember that the final decision is the editor).

Author Response

Thank you very much for your comments.

Reviewer 3 Report

This is an interesting article which was aimed to examinating the validity and reliability of the side bridge test in assessing trunk lateral flexor endurance in recreational female athletes. 

The theme of the article is relevant because still little is known about usefulness (validity and reliability) of the field test in assessing musculoskeletal fitness. In this case - trunk muscle capacity of the physically active young women. 

Whilst the article is interesting, several points are worth addressing.

Minor comments:

Abstract

Line 11 - I suggest to complete information about participants (how many women were examined, mean and sd of the age).

Line 15 - Information that ICC 3.1 was calculated should be suplemented.

Material and methods

Participants

Line 88 - Authors presented the level of the physical activity. What was the method of assessing the PA. It should be described.

Statistical analysis

Line 189 - Authors used the Kolmogorov-Smirnov test to verify normality of the data distribution. My suggestion is to use Shapiro-Wilk test which is more appriopriate for small groups (<50 samples). 

Conclusions

Line 388 - In my opinion Authors should extend the recommendations for comparisons of SBT test results between groups, in relation to the significant correlation with body weight. E.g. using regression analyses adjusted for body weight or ANCOVA with body weight as a confounding variable to compare groups, e.t.c.

Results, Disscussion and References are well described.

Author Response

We would like to thank you for your advice and recommendations, which in our view have contributed to improving the paper. As you will see, we have performed a revision of this manuscript based on your suggestions.

Below we respond to each comment on a point-by-point basis. The authors’ responses to each comment are in bold,

Abstract

Line 11 - I suggest to complete information about participants (how many women were examined, mean and sd of the age).

Response: This information has been included

Line 15 - Information that ICC 3.1 was calculated should be suplemented.

Response: This information has been included

Material and methods

Participants

Line 88 - Authors presented the level of the physical activity. What was the method of assessing the PA. It should be described.

Response: In line 130 (in the procedures section) we mention that the participants filled out a questionnaire about their sport practice but we only asked about frequency, duration, and type of sport practice.

Line 132: “In the first session, the participants filled out a questionnaire their medical history and sport practice in order to know their health status and level of physical activity”

Statistical analysis

Line 189 - Authors used the Kolmogorov-Smirnov test to verify normality of the data distribution. My suggestion is to use Shapiro-Wilk test which is more appriopriate for small groups (<50 samples). 

Response: The Shapiro-Wilk test to verify the normality of the data distribution has been included in the statistical analysis.

Conclusions

Line 388 - In my opinion Authors should extend the recommendations for comparisons of SBT test results between groups, in relation to the significant correlation with body weight. E.g. using regression analyses adjusted for body weight or ANCOVA with body weight as a confounding variable to compare groups, e.t.c.

Response: We have extended the recommendations for comparisons of the SBT test results between groups in relation to the significant correlation with body weight.

Results, Disscussion and References are well described.

Reviewer 4 Report

The authors intended to evaluate the reliability and validity of a side bridge test. Although the idea might be interesting, there are several methodological issues which limit the consideration of the study. 

For reliability, the authors used a very old reference and the cut-offs do not seem appropriate to define the strength of reliability. The definition of >.75 as excellent is not appropriate. It is suggested tor refer to different ranges even more restricting. Moreover, the interpretation provided to explain the difference between the 2 sessions limits a lot the quality of the entire research and study design. 

For validity, considering the theoretical definition, the proposed method/measurement should be evaluated against a gold standard measurement. Authors proposed the assessment of muscles recruitment during the test to assess the validity of the side bridge test. It is suggested to reconsider the approach to the problem. 

The characteristics of the sample are not clear. Authors referred to "physically active" females in methods section and then to "recreational athletes" in discussion section. 

The quality of figures is very low and not scientifically-based. 

From the current study, I would not infer that the test is reliable.

Author Response

We would like to thank you for your advice and recommendations, which in our view have contributed to improving the paper. As you will see, we have performed a revision of this manuscript based on your suggestions.

Below we respond to each comment on a point-by-point basis. The authors’ responses to each comment are in bold,

For reliability, the authors used a very old reference and the cut-offs do not seem appropriate to define the strength of reliability. The definition of >.75 as excellent is not appropriate. It is suggested tor refer to different ranges even more restricting. Moreover, the interpretation provided to explain the difference between the 2 sessions limits a lot the quality of the entire research and study design. 

Response: We have included a new reference with different ranges which are more restricting.

“The interpretation of the ICC was made based on the following values: excellent (0.90-1.00), good (0.70-0.89), moderate (0.50-0.69) and low (<0.50)”

Munro, B. H. (2005). Statistical Methods for Health Care Research (5th ed.). Philadelphia: Lippincott Williams & Wilkins.

Regarding the difference between the two sessions, we were also surprised by the result.  We did not find other interpretation to explain the difference between the 2 sessions. However, we think that this circumstance does not limit the quality of the research since the ICC was good (0.81) and the correlation analysis was not affected by this result.  

For validity, considering the theoretical definition, the proposed method/measurement should be evaluated against a gold standard measurement. Authors proposed the assessment of muscles recruitment during the test to assess the validity of the side bridge test. It is suggested to reconsider the approach to the problem. 

Response: There are several types of validity. We did not analyze the concurrent validity which has to be evaluated against a gold standard measurement as you indicate.

Muscle recruitment has been previously used to assess the validity of similar trunk endurance tests until exhaustion (De Blaiser et al., 2018; Coorevits et al., 2008). Electromyographic (EMG) spectrum analysis has been generally used to monitor the development of localized muscle fatigue, because fatigue causes a decrease of the frequency content of the EMG signal, usually described as a decline of the median frequency parameters of the EMG spectrum. Furthermore, it has been proven that local muscle endurance is associated with fatigue-based changes in EMG properties. Therefore, one way to demonstrate the validity of the SBT for measuring could be to show which muscles are fatigued the most during the test.

De Blaiser C, De Ridder R, Willems T, Danneels L, Vanden Bossche L, Palmans T, et al. Evaluating abdominal core muscle fatigue: Assessment of the validity and reliability of the prone bridging test. Scand J Med Sci Sport. 2018 Feb 1;28(2):391–9.

Coorevits P, Danneels L, Cambier D, Ramon H, Vanderstraeten G. Assessment of the validity of the Biering-Sørensen test for measuring back muscle fatigue based on EMG median frequency characteristics of back and hip muscles. J Electromyogr Kinesiol. 2008 Dec;18(6):997–1005.

The characteristics of the sample are not clear. Authors referred to "physically active" females in methods section and then to "recreational athletes" in discussion section. 

Response: The description of the sample characteristics in the discussion section has been changed.

The quality of figures is very low and not scientifically-based.

Response: We have improved the quality of the figures, which are very similar to those used in previous studies which analyzed similar trunk endurance tests published in the Scandinavian Journal of Medicine and Science in Sport and in the Journal of Electromyography Kinesiology.

De Blaiser C, De Ridder R, Willems T, Danneels L, Vanden Bossche L, Palmans T, et al. Evaluating abdominal core muscle fatigue: Assessment of the validity and reliability of the prone bridging test. Scand J Med Sci Sport. 2018 Feb 1;28(2):391–9.

Coorevits P, Danneels L, Cambier D, Ramon H, Vanderstraeten G. Assessment of the validity of the Biering-Sørensen test for measuring back muscle fatigue based on EMG median frequency characteristics of back and hip muscles. J Electromyogr Kinesiol. 2008 Dec;18(6):997–1005.

From the current study, I would not infer that the test is reliable.

Response: The SBT showed a good relative reliability with an ICC= 0.81, which is similar to previous studies. Regarding the absolute reliability, unfortunately, for many tests in sports science (especially those tests until exhaustion) the SEM is very high. Whether a test's level of precision is acceptable, therefore, may depend on the intended use of the test as we indicated in the conclusions.

Evans K, Refshauge KM, Adams R. Trunk muscle endurance tests: reliability, and gender differences in athletes. J Sci Med Sport. 2007 Dec;10(6):447-55. doi: 10.1016/j.jsams.2006.09.003. Epub 2006 Dec 1. PMID: 17141568.

Round 2

Reviewer 4 Report

Although the effort of authors in revising the manuscript, the quality of study and manuscript remain still low in light of the journal level. 

Author Response

Dear Reviewer,

We appreciate your opinion very much.

The manuscript has been sent to Proof-Reading-Service.com for editing and proofreading, as can be seen in the attached certificate.

As indicated in previous revision, this study has followed a study design very similar to others conducted on similar endurance tests (i.e. Biering-Sorensen test, prone bridging test) which have been published in other Q1 JCR journals.
